# Nanographenes as electron-deficient cores of donor-acceptor systems

Yu-Min Liu[1], Hao Hou[1], Yan-Zhen Zhou[1], Xin-Jing Zhao[1], Chun Tang[1], Yuan-Zhi Tan [1] & Klaus Müllen[2]

Conjugation of nanographenes (NGs) with electro-active molecules can establish donor-acceptor π-systems in which the former generally serve as the electron-donating moieties due to their electronic-rich nature. In contrast, here we report a series of reversed donor-acceptor structures are obtained by C–N coupling of electron-deficient perchlorinated NGs with electron-rich anilines. Selective amination at the vertexes of the NGs is unambiguously shown through X-ray crystallography. By varying the donating ability of the anilino groups, the optical and assembly properties of donor-acceptor NGs can be finely modulated. The electron-deficient concave core of the resulting conjugates can host electron-rich guest molecules by intermolecular donor-acceptor interactions and gives rise to charge-transfer supramolecular architectures.

[1] Collaborative Innovation Center of Chemistry for Energy Materials, State Key Laboratory for Physical Chemistry of Solid Surfaces, and Department of Chemistry, College of Chemistry and Chemical Engineering, Xiamen University, 361005 Xiamen, China. [2] Max Planck Institute for Polymer Research, Ackermannweg 10, D-55128 Mainz, Germany. These authors contributed equally: Yu-Min Liu, Hao Hou. Correspondence and requests for materials should be addressed to Y.-Z.T. (email: yuanzhi_tan@xmu.edu.cn)

As molecularly defined cutouts of the graphene lattice, nanographenes (NGs) have been developed to deduce intrinsic structure-property correlations[1–3]. Among various functional characteristics of NGs, their extended $sp^2$-conjugated carbon skeleton offers intriguing optical absorption, photoluminescence and charge transport properties[4–7] and thus promising potential in electronics[8, 9] and optoelectronics[10]. The electron-rich nature of NGs qualifies them as typical p-type semiconductors in organic field-effect transistors[11–14] and as electronic donors in organic photovoltaics[14–16]. Allowing conjugation of donor to acceptor moieties has proven as a powerful concept to modulate the optical absorption and charge transport[17–22]. In such donor-acceptor (D-A) conjugates[23–25], NGs have typically served as donors. Recently, a few studies have switched the electronic characteristics of NGs from electron-rich to electron-poor by multiple introduction of strongly electron-withdrawing groups, such as bisimide[26–29], chloro[30], and fluoro[31, 32] substituents. Such electron-deficient NGs thus provide alternative pathway to D-A conjugates. However, this concept remains challenging due to lack of suitable synthetic protocols.

Here we report an amination occurring at the vertexes of perchlorinated NGs by palladium-catalyzed Buchwald–Hartwig coupling[33, 34]. Using this strategy, various electron-rich aniline derivatives are coupled to the periphery of electron-deficient chlorinated NGs, providing a series of nano-sized D-A architectures. The concave molecular structures of these D-A NGs are clearly demonstrated by single-crystal X-ray diffraction, which show a selective amination at the vertexes. The donating strength of attached anilino groups can finely tune the optical absorption and intermolecular interactions. Different from previously reported concave polycyclic aromatic hydrocarbons, the concave electron-deficient skeletons of the D-A NGs can assemble with donor-type guest molecules such as tetrathiafulvalene by intermolecular charge transfer (CT).

## Results

**Amination of NGs.** Arm-chair edges of graphenic structures comprise two types of carbon sites[35–37], one at the bay and another at the vertex (Fig. 1). The steric hindrance at the bay position is obviously larger than that at the vertex and will become even more severe with bulky groups. In case of perchlorinated NGs, the strong congestion of chloro substituents at the bay has been shown to distort the molecular geometry from planar to curved[30]. We reasoned that the chlorines at the different peripheral sites of NGs possessed different chemical reactivity and could allow selective modification (Fig. 1).

Taking perchlorinated hexa-peri-hexabenzocoronene ($C_{42}Cl_{18}$, **1**, here the smallest representative NG) with two types of chloro groups as an initial example, we screened the palladium-catalyzed

Buchwald–Hartwig C–N cross-coupling between **1** and aniline (Fig. 2 and Supplementary Table 1). Under optimized reaction conditions aniline can, indeed, dominantly couple to the vertexes of **1**, yielding the hexakis-anilino chlorinated hexa-peri-hexabenzocoronene (**2a**) (Fig. 2 and Supplementary Fig. 1). The bulky bidentate phosphine ligand was necessary to produce **2a** (Supplementary Table 1).

In addition to NMR spectroscopy (Supplementary Figs. 9 and 10), the structure of **2a** was assessed by single crystal X-ray diffraction (Fig. 2b). As depicted in Fig. 2b, the most distinguishable feature of **2a** is six substituting anilino groups at the vertexes of NG. The unobstructed remaining chlorines in the bays adopt an up and down conformation, which keeps the structure of **2a** doubly concave (Fig. 2). We further demonstrated that the amination was tolerant to different aniline derivatives, because **2b-2e** were synthesized and characterized (Fig. 2 and Supplementary Figs. 4, 7 and 11–18).

**Optical properties of 2.** The UV-Vis spectrum of **2a** displays three bands that peak at 452, 482, and 560 nm (Fig. 2c). The variation of concentration does not affect the absorption profile of **2a**, suggesting the absence of the $\pi-\pi$ stacking in solution (Supplementary Fig. 36). The absorption of **2a** showed a bathochromic shift (45 nm) compared with its parent compound **1**[30], attributed to the intramolecular CT from anilino groups to the inner NG. An expected solvatochromism with increasing solvent polarity was observed for **2a**, but the redshift is relatively small (Supplementary Fig. 35).

Comparing the optical absorption, a gradual bathochromic shift of absorption peaks was revealed (Fig. 2c and Supplementary Table 2) from **2a** to **2e** without concentration dependence (Supplementary Figs. 36–40), in which the electron-donating power of peripheral anilino groups increased. Consequently, the optical HOMO-LUMO gap of **2a** to **2e** decreased from 2.18 to 1.84 eV (Supplementary Table 2). On the other hand, the photoluminescence of compounds **2a**, **2b**, and **2c** shows an emission with $\lambda_{max}$ located at ~560 nm with an absolute photoluminescence quantum yield (PLQY) of 3.5–5.4% (Supplementary Table 2), whereas the enhanced intramolecular CT in **2d** and **2e** quenched their photoluminescence dramatically (Supplementary Fig. 41 and Supplementary Table 2). These optical properties of **2** confirmed their D–A characteristics.

**Theoretical calculations of 2.** Theoretical calculations helped to describe the intramolecular CT in **2**. LUMOs were generally distributed over the NG moiety whereas the HOMOs were located at the peripheral anilino units (Supplementary Fig. 52). The electron density differences between the first excitation and the ground state showed a decrease at the anilino units and an increase at the inner NG core (Supplementary Fig. 53) in **2**. Differently, the weaker donating ability of anilino groups in **2a** leads to poor intramolecular CT, therefore the electron density decrease at the anilino groups of **2a** is obviously smaller than that in **2e** (Supplementary Fig. 53), which comprises the electron-richest functional groups. Theoretical and experimental results thus jointly validate the D-A nature of **2**.

**Packing structure of 2.** The crystal structures of **2a-2e** (Supplementary Figs. 31–33 and Supplementary Methods) revealed their assembly in the solid state. All inner cores of **2a-2e** have a $C_{3v}$ symmetrical double concave structure, if ignoring the conformation of peripheral phenyl groups (Fig. 2b and Supplementary Figs. 31–33), which impedes the $\pi-\pi$ interactions between the NG cores in the solid (Supplementary Fig. 42). Indeed, the solid-state UV-Vis spectra of **2a–2d** display small

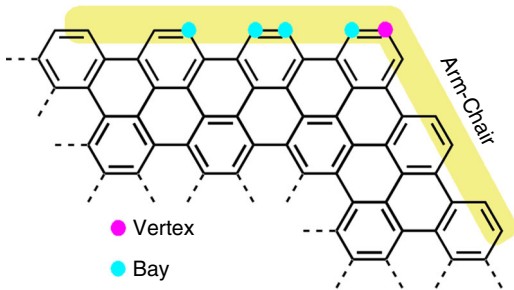

**Fig. 1** Arm-chair edge of graphenic structures. Arm-chair edge contains two kinds of carbon sites, one at the bay (cyan point) and another at the vertex (magenta point)

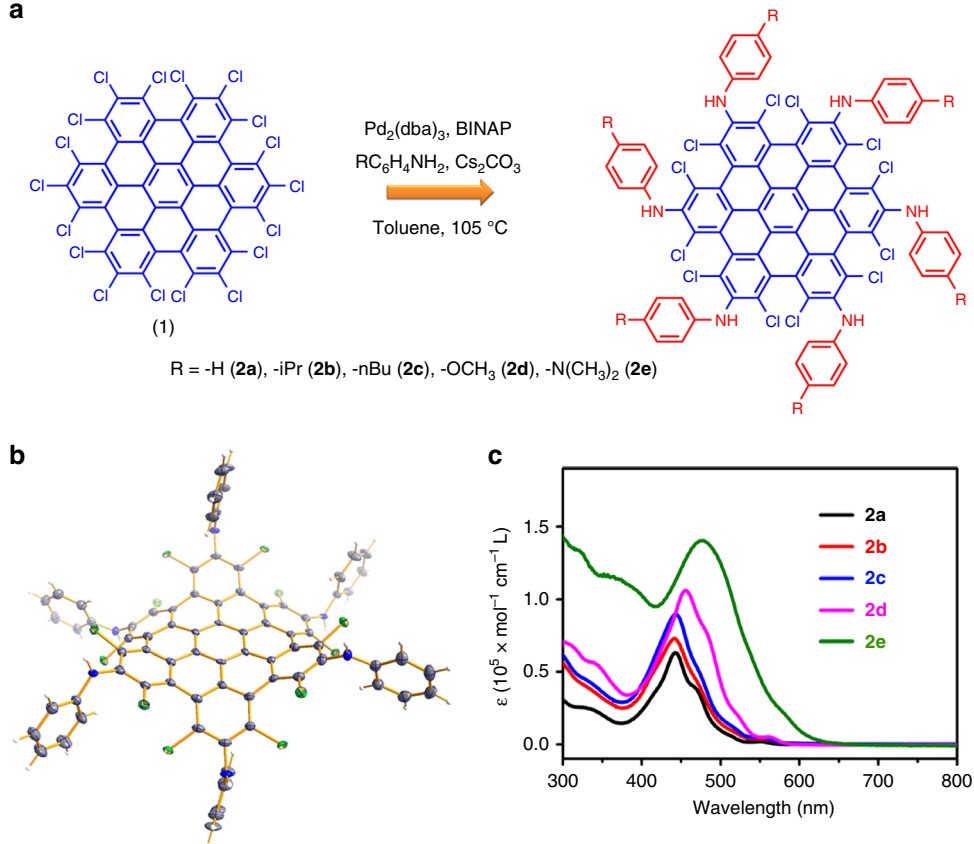

**Fig. 2** Synthesis and crystal structure of **2** and optical properties. **a** optimized synthetic route for **2**. **b** the doubly concave structure of **2** represented by **2a**. The thermal ellipsoids are set at 50% probability level. **c** UV-Vis absorption spectra of **2** measured in dichloromethane (the concentration of **2** is $10^{-5}$ mol·L$^{-1}$). Pd$_2$(dba)$_3$ = tris(dibenzylideneacetone)dipalladium. BINAP = 2,2′-bis(diphenylphosphino)-1,1′-binaphthalene

red shift by 30–35 nm (Supplementary Figs. 43–46 and Supplementary Methods), in comparison with those in solution, which confirms the absence of π–π interactions in the solid state for **2a**-**2d**.

In contrast, in the crystal of **2e**, the peripheral p-N-N-dimethyl anilino groups assemble with the skeleton of NG core in a face-to-face fashion with a π–π interaction distance of 3.30 Å (Fig. 3), forming intermolecular D–A supramolecular networks. The different assembly structure of **2e** originates from the stronger donor ability of p-N-N-dimethyl anilino groups, as compared with those in **2a**-**2d**. Remarkably, an obvious red shift by 125 nm for **2e** appeared in the solid-state optical absorption (Fig. 3), and the absorption of **2e** in the solid extends over the entire range of visible light. The case of **2e** suggests that the intermolecular CT creates an extensive absorption of the D–A molecules, which should be useful for the design of conjugated molecules for light harvesting[38].

**Amination for larger NGs**. Encouraged by the successful amination of **1**, we proceeded to larger homologs of **1**, C$_{60}$Cl$_{22}$ (**3**) and C$_{78}$Cl$_{26}$ (**5**), which contain six chlorines at the vertexes as well. Under similar reaction conditions, two series of hexakis-aminated chlorinated products **4a**–**4c** (C$_{60}$Cl$_{16}$(NHC$_6$H$_4$R)$_6$, R = −H (**4a**), −nBu (**4b**), and −OCH$_3$ (**4c**)) and **6a**–**6b** (C$_{78}$Cl$_{20}$(NHC$_6$H$_4$R′)$_6$, R′ = −nBu (**6a**) and −OCH$_3$ (**6b**)) were obtained (Fig. 4 and Supplementary Figs. 2, 5, 6, 8 and 19–30). The unambiguous structure of **4** was disclosed by single-crystal X-ray diffraction (Supplementary Methods). The Fig. 4b showed

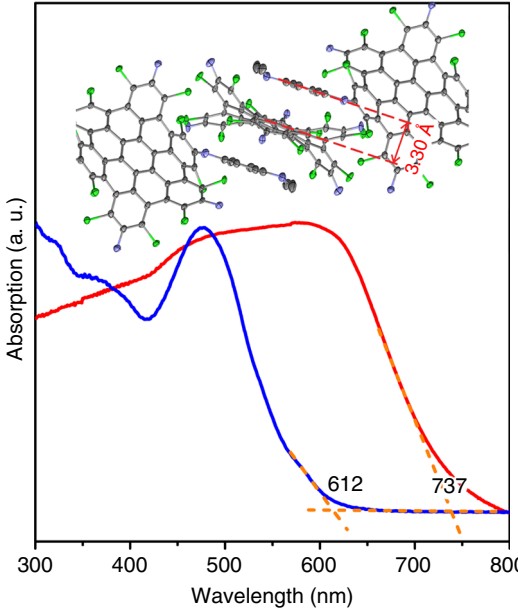

**Fig. 3** Absorption and assembly in the solid state of **2e**. The UV-Vis absorption of **2e** in solid state (red) measured in a diffuse-reflectance mode was compared with that in dichloromethane (blue, $10^{-5}$ mol·L$^{-1}$). The peripheral groups which do not allow D–A interactions with the inner NG were omitted for clarity

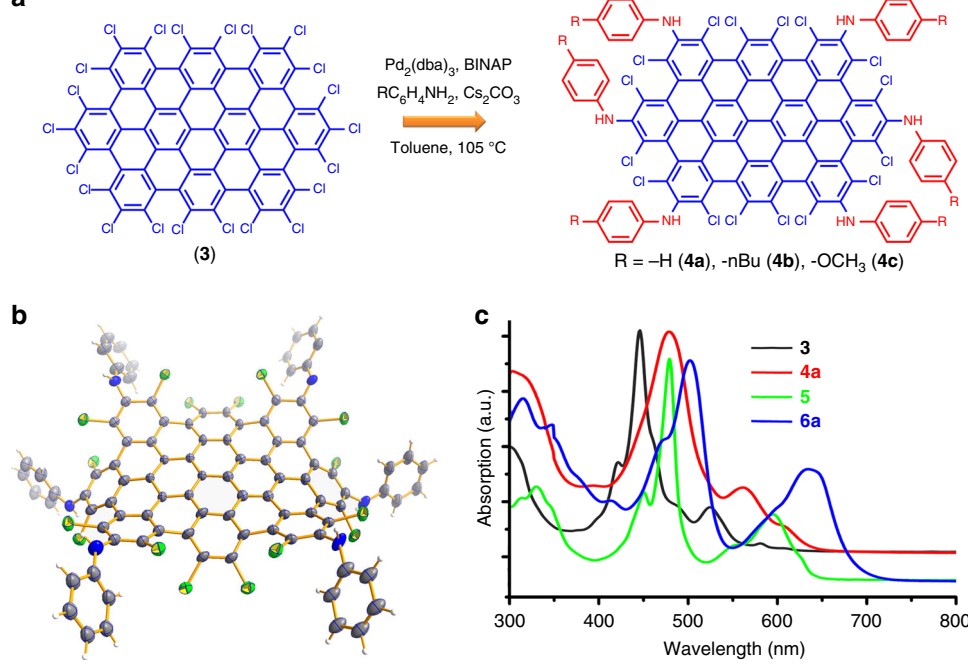

**Fig. 4** Synthesis and crystal of **4** and optical absorption. **a** optimized synthetic route for **4**. **b** the crystal structure of **4** represented by **4a**. The thermal ellipsoids were set at 30% probability level. **c** UV-Vis absorption spectra of **3**, **4a**, **5**, and **6a**. Notably, the molar absorption coefficients of **3**, **4**, and **6** were listed in Supplementary Figs. 47–49, while that of **5** can not be measured due to its extremely low solubility

that six anilino groups are indeed connected to the vertexes of **3** while the chlorines at the bay again keep intact. The UV-Vis spectra of **4** and **6** display bathochromic shifts by 39 and 37 nm (Fig. 4c, Supplementary Figs. 47–49 and Supplementary Table 3), respectively, compared with **3** and **5**, also due to the intramolecular CT. By theoretical calculations, **4** exhibits a typical distribution of electron density differences as a feature of D-A conjugates (Supplementary Fig. 55).

**Supramolecular assembly between D–A NGs and tetrathiafulvalene.** D–A NGs own an electron-deficient concave core, which can act as supramolecular host for electron-rich donor molecules such as tetrathiafulvalene (TTF). Evaporation of the carbon disulfide solution of **2a** and TTF led to the formation of crystalline black needles, different from the orange prisms of **2a**. X-ray diffraction analysis revealed that a supramolecular complex **2a** ⊃ TTF had formed with TTF molecules located at the center of concave NG in a face-to-face manner (Fig. 5a, c). The interfacial distance between TTF and NG core is 3.30 Å implying strong π–π interactions. Further stacking between TTF and **2a** builds up a mixed D–A–D–A supramolecular column (Fig. 5c). The columnar supramolecular architecture of **2a** ⊃ TTF holds promise for long-range orientations of CT dipoles, making it a potential organic ferroelectric[39, 40]. Compound **4** as acceptor can also host electron-donating TTF (Fig. 5 and Supplementary Fig. 34). One TTF molecule is located above the core of **4**, forming sandwich-type complexes (2·**4** ⊃ TTF) with close interfacial π–π interactions (3.30 Å) between TTF and inner NG.

The structures of both supramolecular complexes (**2a** ⊃ TTF and 2·**4** ⊃ TTF) suggest the presence of intermolecular CT between TTF and electronic deficient core as proven by CT bands in the absorption spectra of **2a** ⊃ TTF and 2·**4** ⊃ TTF (Fig. 5). In comparison with **2a** ⊃ TTF, a bathochromic shift (130 nm) appeared in the CT band of 2·**4** ⊃ TTF (Fig. 5f and

Supplementary Fig. 50), attributed to the extended conjugated skeleton of **4**. Remarkably, the absorption of 2·**4** ⊃ TTF crystals can extend into the NIR region up to 1000 nm. The unambiguous structure and clear CT band of the supramolecular complexes between D–A NGs and donating guests (Fig. 5) characterized these D–A NG conjugates as promising electronic-accepting host for the construction of CT supramolecular architectures.

## Discussion

Our amination protocol enabled functionalization at the vertexes of NGs and constructed reversed D-A conjugates. The donating ability of aniline groups affords an efficient approach to modulate the properties of D-A NGs, e.g., absorption, optical gap, luminescence, and assembly. Since the edge halogenation of graphene nanoribbons and graphenic materials was achieved, our functionalization concept can additionally incorporate D-A structures and CT dipoles into these graphenic structures. The resulting D–A NGs undergo assembly with electron-rich molecules by intermolecular D-A interactions, creating CT supramolecular architectures. This opens an avenue toward supramolecular functional materials based on NG acceptors.

## Methods

**Synthesis of 2.** First, the reaction conditions, including amount of aniline, phosphine ligands and bases, for the palladium-catalyzed Buchwald–Hartwig C-N coupling between **1** and aniline were screened (Supplementary Table 1). The optimized conditions required a large excess of aniline to favor the multiple C–N coupling of **1** with aniline and suppress the homocoupling between **1**[41, 42], supported by the control experiments (Supplementary Table 1). The decreased amount of aniline in control experiments resulted in a lower yield of **2a**, whereas the experiments with 15 and 30 fold excess of aniline did not affect the yield (Supplementary Table 1).

Under the optimized reaction conditions, a 25 ml reaction tube was charged with **1** (100 mg, 0.087 mmol), $Cs_2CO_3$ (113 mg, 0.348 mmol), $Pd_2(dba)_3$, (24 mg, 0.026 mmol) and BINAP (32 mg, 0.052 mmol) under argon, and then an aniline (194 mg,

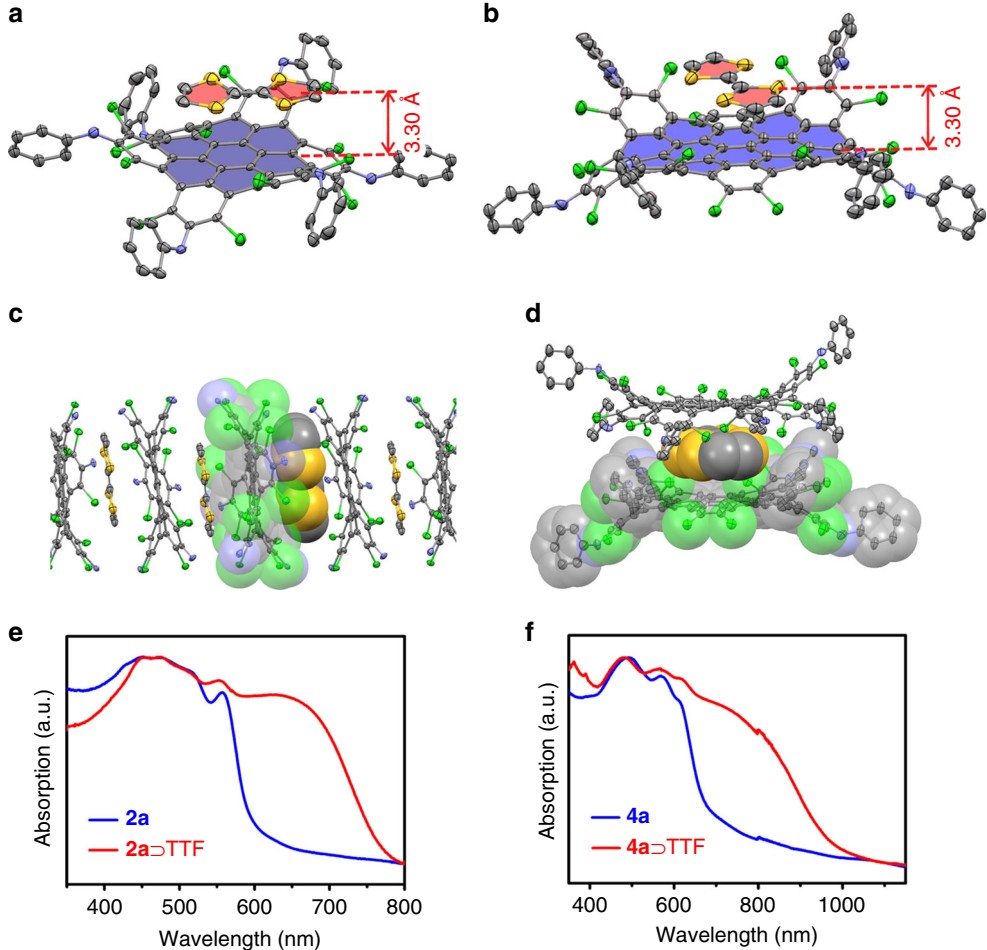

**Fig. 5** Supramolecular assembly between D-A NGs and TTF. **a**, **b** the structures of **2a** ⊃ TTF and **4a** ⊃ TTF. **c** one dimensional mixed-stacking of **2a** and TTF in the crystal. Peripheral phenyl groups were omitted for clarity. **d** sandwich-type complex of 2·**4a** ⊃ TTF. **e**, **f** UV-Vis-NIR spectra of **2a**, **2a** ⊃ TTF, **4a** and 2·**4a** ⊃ TTF in the solid state measured in a diffuse-reflectance mode. CT bands at 650 nm and 780 nm were clearly observed in the spectra of supramolecular complex

2.09 mmol, equal to 4-fold according to the molar of chlorine at the vertexes) solution in toluene (5 ml) was added into the tube. The reactants were stirred and refluxed at 105 °C for 36 h. After the reaction, the products were extracted with dichloromethane (DCM) (30 ml) and washed with water (20 ml). The collected organic phase was dried by anhydrous $MgSO_4$. After filtration and evaporation of the solvent, the crude products were separated over a silica column using DCM/ petroleum ether (60–90 °C) (1.5:1) as eluent. The red component was collected, yielding 40 mg of **2a**. The yield was 31% (Note that the yield of **2a** determined by NMR spectroscopy is ~50%, Supplementary Table 1). Due to strong absorption of **2a** on silica, the subsequent eluent contained **2a** as well (Supplementary Fig. 1). If these components (spot 3 shown in Supplementary Fig. 1) were collected and separated again, the isolated yield can be improved to 38%.

Besides **2a** as the major product of C–N coupling, the byproducts in the reaction were attributed to the oligomers originating from the competitive palladium-catalyzed C–C homocoupling between **1**[41, 42] and a few pentakis-anilino products. The byproducts with more than 6 anilino groups were not observed. Considering that six C-N bonds are formed in one reaction, the achieved yield is over 80% for each C–N bond formation.

**2b**, **2c**, **2d**, and **2e** were synthesized by the coupling of **1** to different aniline derivatives employing similar reaction and separation conditions. The isolated yields of **2b**, **2c**, **2d**, and **2e** were 32, 26, 26, and 21%, respectively.

**Synthesis of 4**. A 25 ml reaction tube was charged with **3** (50 mg, 0.033 mmol), $Cs_2CO_3$ (43 mg, 0.133 mmol), $Pd_2(dba)_3$ (9 mg, 0.01 mmol) and BINAP (12 mg, 0.02 mmol) under argon. An aniline (74 mg, 0.792 mmol, equal to 4-fold according to the molar of chlorine at the vertexes of **3**) solution in toluene (5 ml) was added into the tube. After stirring at 105 °C for 36 h, 30 ml DCM was added. The organic phase was washed with water (20 ml) and dried over anhydrous $MgSO_4$. After removing the solvent, the crude products were separated by a silica column using DCM/petroleum ether (60–90 °C) (1.5:1) as eluent. The collected component was

further purified by high performance liquid chromatography (HPLC) using JAIGEL-2H column (Japan Analytical Industry Ltd., chloroform as eluent), yielding 10 mg of **4a**. The yield was 17%.

**4b**, and **4c** were also synthesized using different aniline derivatives under similar reaction and separation conditions as those of **4a**. Notably, purification by HPLC using JAIGEL-2H column was necessary for **4b** and **4c** as well. The isolated yields of **4b** and **4e** were 13 and 20%, respectively, under the optimized conditions.

**Synthesis of 6**. **5** (50 mg, 0.027 mmol), $Cs_2CO_3$ (36 mg, 0.133 mmol), $Pd_2(dba)_3$ (8 mg, 0.0099 mmol), and BINAP (10 mg, 0.0198 mmol) were introduced into a 25 ml reaction tube under argon (Supplementary Fig. 2). Then toluene (5.0 ml) and 4-n-butyl-aniline (96 mg, 0.792 mmol, equal to a 4-fold according to the molar of chlorine at the vertexes) were added. The reactants were stirred for 36 h at 105 °C. The products were extracted with DCM (30 ml), washed with water (20 ml) and dried over anhydrous $MgSO_4$. After removing the solvent, the crude products were separated over a silica column using DCM/petroleum ether (60–90 °C) (1:3) as the eluent. Then, the obtained component was further purified by HPLC using JAIGEL-2H column (chloroform as the eluent), yielding 13 mg of **6a**. The yield was 19%. Similarly, **6b** was synthesized and isolated, with a yield of 8%.

**Crystallography**. Using Olex2[43], all the initial structures were solved with the SHELX-XT structure solution program by the direct method and refined with the XL refinement package by Least Squares minimization. The crystals of **4a** and **4c** easily effloresced, thus the structures of **4a** and **4c** were obtained from the crystal structures of their respective supramolecular complexes. For compound **6**, single crystals could be obtained which were too tiny to be measured by X-ray diffraction.

**Theoretical calculations**. All the calculations were performed with the Gaussian 09 software package[44]. The solvent effects of DCM used in experiment have been

considered using the CPCM continuum solvation model[45]. First, in order to ensure the suitable hybrid functional, we compared the theoretical HOMO–LUMO gap calculated by B3LYP, CAM-B3LYP, M062X, and HSEH1PBE[46] with the experimental optical HOMO–LUMO gap (Supplementary Table 4). The optimized structures by different functionals were also compared with the experimental crystallographic data, taking **2a** as an example (Supplementary Fig. 51, Supplementary Table 5). Accordingly, the HSEH1PBE functional is more appropriate for the theoretical calculations (Supplementary Figs. 52–55). Then we calculated the electron density differences between the first excitation state (corresponding to the UV-Vis absorption with maximum wavelength and mainly arising from HOMO to LUMO transition) and the ground state for compounds **2–4** by HSEH1PBE.

**Data availability**. Supplementary crystallographic data for this manuscript have been deposited at the Cambridge Crystallographic Data Centre under deposition numbers CCDC 1580856–1580862. These data can be obtained free of charge from [http://www.ccdc.cam.ac.uk/data_request/cif]. The authors declare that all the data supporting the findings of this study are available within the article (and Supplementary Information File), or available from the corresponding author on reasonable request.

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

## Acknowledgements

This work was financially supported by the National Natural Science Foundation of China (21771155, 21721001), the Ministry of Science and Technology of China (2014CB845603, 2017YFA0204902), the Thousand Youth Talents Plan and the Fundamental Research Funds for the Central Universities (20720180035). We appreciated Prof. Zexing Cao and Mr. Mingjun Sun for the help on the theoretical calculations.

## Author contributions

K.M. and Y-Z.T. conceived and designed the experiments; Y.-M.L, H.H, Y.-Z.Z., and X.-J.Z. conducted synthesis and completed the identification; C.T. performed the theoretical

work; Y.-M.L., Y.-Z.T., and K.M. co-wrote the paper; All authors discussed the results and commented on the manuscript.

## Additional information

**Competing interests:** The authors declare no competing interests.

