## [Peer Review File · Nature Communications]

Reviewers' comments:

Reviewer #1 (Remarks to the Author):

In the manuscript the authors describe regioselective amination of perchlorinated large polyaromatic hydrocarbons (nanographenes, NGs) by palladium-catalyzed Buchwald–Hartwig coupling, which selectively occurred at the vertexes of NGs. The authors well chemically and structurally characterized the obtained derivatives and probed their electronic properties. They present a suitable protocol for modulating electron-donating properties of the functionalized NGs. The manuscript is clearly and well written. However, the modulation of electronic properties of NGs by functionalization was already published and therefore the reviewer feels the manuscript is more suitable for specialized chemical journal.

Specific points:

Figure 1c – x-axis should be labeled, solvent should be specified.

It is hard to agree with the claims from DFT calculations aimed at providing insight into intramolecular CT (pg. 6). Analysis based on calculations of ground/excited states and electron density differences maps would provide more insightful information. In addition, an order of orbitals strongly depends on the choice of DFT functional and various functionals may provide different frontier orbitals. How was this issue handled by the authors?

Figure 2 – How was the “Absorption” defined? Details about acquiring solid state UV/Vis should be provided at least in ESI.

Reviewer #2 (Remarks to the Author):

This manuscript has presented synthesis of a series of Donor-Acceptor molecular architectures by amination of perchlorinated nanographenes with anilines, and structures and properties of these molecules, as well as the complexes of these molecules with TTF exhibiting intermolecular charge transfer. These D-A molecular architectures are interesting functional π -molecules with potential

applications as organic electronic materials and functional dyes. However, two key findings as claimed in this manuscript have not been supported with solid experimental evidences.

1) The reactions of perchlorinated nanographenes 1, 3 and 5 with anilines by palladium-catalyzed Buchwald–Hartwig are claimed as regioselective, and only the products that have the vertex chlorine atoms substituted by nitrogen were reported. However, the yields of compounds 2a-e are 26-32%, the yields of 4a-c are 13-20%, and the yields of 6a-b are as low as 8-19%. What are the other products (68% to 92%) of these reactions? Did amination always occur on the vertex chlorines?

2) The structures of compounds 2a, 2c, 2d, 2e, 4a and 4c were unambiguously characterized by X-ray crystallography. It is reasonable to establish the structures of 2b and 4b on the basis of these crystal structures and comparison of ¹H NMR spectra of 2b and 4b with those of 2a, 2c, 2d, 2e, 4a and 4c. However, the claimed structures of 6a and 6b lack solid support from experimental results. On Page 5 of the Supporting Information, the authors noted the difficulties in conducting elemental analysis and mass spectra. However, without molecular formula as determined by elemental analysis or mass spectra, one cannot even know how many chlorine atoms in compound 5 were substituted with anilines. Without appropriate experimental evidences, the structures of 6a and 6b should not be claimed.

Moreover, some typos and grammar mistakes are found in the manuscript, and the writing should be improved. For example, in Line 34 (Page 2), "a few of" should be "a few", and in Line 78 (Page 5), "substituted" should be "substituting". And the exact meaning of the sentence "The unobstructed chlorines in the bays are staggered" in Line 79 (Page 5) is not clear.

Therefore, this referee does not recommend this manuscript in the current form to be published in Nature Communications. But a revised manuscript with the above issues addressed appropriately can be reconsidered.

Reviewer #3 (Remarks to the Author):

The authors had previously developed a method to perchlorinate HBCs. In this work, they showed regioselective amination at the vertexes of the perchlorinated HBCs. Taking advantage of the

reactivity difference of Cl at the vertex and bay sites, they were able to replace all vertex Cl in reasonable yield. They have also shown that the perchlorinated HBCs have an electron-deficient concave core, which could form charge-transfer supramolecular structures with electron-rich donor like TTF. This review will primarily focus on the chemistry part of the work. The manuscript can be accepted for publication, provided that the evaluation on the charge transfer part of the work is deemed significant.

The regioselectivity demonstrated in this work is quite interesting. It takes advantage of the difference in the steric environment to modulate the reactivity of Cl at the vertex and bay sites in the C-N cross-coupling reaction. The identified reagents and reaction conditions allowed them to completely replace 6 vertex chlorine with aniline. This work will be of interest to synthetic organic chemists and materials scientists working on carbon materials.

How general is this method? The 3 examples have the same number of "layers", which gave 6 vertex sites for substitution. What are the challenges when the structures are larger?

In the rest of the ~70% product, what is the distribution of the various substituted products?

The charge transfer complex was done on the H or alkyl substituted aniline. How would other substituents on aniline (eg. OCH₃) impact the complex formation and their properties?

Several different terminologies were used to describe their materials, for example, HBC, nanographene, larger graphene, graphene nanoribbons. One should be very careful choosing the correct description of these materials, as the term "graphene" refers to a very special type of material having unique physical properties.

The reaction conditions in Figures 1 and 3 are missing the critical aniline compound.

The quality of the spectra could be improved. Several spectra have solvent peaks, silicone grease in addition to water peaks.

Compound labels are missing in several figures. For example, compound 1 in Figure 1A, compound 3 in Figure 3A, compound 5 in Figure 3B.

Abbreviations should be defined for the benefit of general audience, for example, Pd2(dba)3, rac-BINAP, TTF.

Reviewers' comments:**Reviewer #1 (Remarks to the Author):**

In the manuscript the authors describe regioselective amination of perchlorinated large polyaromatic hydrocarbons (nanographenes, NGs) by palladium-catalyzed Buchwald–Hartwig coupling, which selectively occurred at the vertexes of NGs. The authors well chemically and structurally characterized the obtained derivatives and probed their electronic properties. They present a suitable protocol for modulating electron-donating properties of the functionalized NGs. The manuscript is clearly and well written. However, the modulation of electronic properties of NGs by functionalization was already published and therefore the reviewer feels the manuscript is more suitable for specialized chemical journal.

Response: We appreciate the reviewer for the valuable comments and suggestions. Our work presented the first case of donor-acceptor (D-A) system built by nanographene (NG) acceptor, which offered an alternative and new way to construct D-A conjugates using NG moiety. Moreover, beyond C-N coupling, the concept of the regioselective modification at the vertex of NGs can be extended, which furnishes a series of functional NGs with electron-accepting cores.

[redacted]

Detailed studies on the selective C-S coupling at the vertexes and concave-convex supramolecular assembly are under way and we would like to report these in due course. These points and the wider impact of this chemical modification render our work significantly different from the reported studies on “the modulation of electronic properties of NGs by functionalization”. Even more so since we describe not only the molecular, but also the supramolecular consequences.

Specific points:

Figure 1c – x-axis should be labeled, solvent should be specified.

Response: Thanks for the comments of reviewer. The x-axis was labeled in Figure 1c and the solvent was specified.

It is hard to agree with the claims from DFT calculations aimed at providing insight into intramolecular CT (pg. 6). Analysis based on calculations of ground/excited states and electron density differences maps would provide more insightful information. In addition, an order of orbitals strongly depends on the choice of DFT functional and various functionals may provide different frontier orbitals. How was this issue handled by the authors?

Response: The excitation states for compounds **2**, **4** and **6** were calculated by time-dependent density-functional theory (TD-DFT). First, in order to ensure the suitable hybrid functional, we compared the theoretical HOMO-LUMO gap (G) of **2c**, **4b**, and **6a** (all of them were decorated with 4-n-butyl-anilino groups at the vertexes) calculated by B3LYP, CAM-B3LYP, M062X and HSEH1PBE with the experimental optical HOMO-LUMO gap ($G_{\text{opt, exp.}}$) (Table 1). The optimized structure by different functionals was also compared with the experimental crystallographic data, taking **2a** as an example (Table 2). Accordingly, the HSEH1PBE functional is more appropriate for the theoretical calculations.

Table 1. Comparison of theoretically calculated G and $G_{\text{opt, exp.}}$.

	$G_{\text{opt, exp.}}$ (eV)	G_{B3LYP} (eV)	$G_{\text{CAM-B3LYP}}$ (eV)	G_{M062X} (eV)	G_{HSEH1PBE} (eV)
2c	2.15	2.73	4.91	4.51	2.33
4b	1.95	2.37	4.43	4.03	1.96
6a	1.72	2.11	4.07	3.69	1.72

Figure 3. Numbering of the carbon atoms of **2a**

Table 2 Comparison of the bond lengths (BL) between experimental data (crystal structure) and theoretically optimized structure of **2a** and relative deviations (RD).

	C ₁ -C _{1'} (Å)	C ₁ -C ₂ (Å)	C ₂ -C ₃ (Å)	C ₃ -C _{3'} (Å)	C ₃ -C ₄ (Å)	C ₄ -C ₅ (Å)	RD (%) ^[a]
Experimental	1.412	1.435	1.420	1.459	1.403	1.402	0
B3LYP	1.415	1.439	1.423	1.471	1.407	1.414	0.44%
CAM-B3LYP	1.406	1.442	1.413	1.470	1.400	1.406	0.44%
M062X	1.409	1.444	1.412	1.469	1.402	1.408	0.43%
HSEH1PBE	1.410	1.434	1.416	1.463	1.404	1.410	0.23%

^[a] RD = $\Sigma[(|BL_{\text{optimized}} - BL_{\text{experimental}}|) / BL_{\text{experimental}}] / n$.

Then we calculated the electron density differences between the first excitation state (corresponding to the UV-Vis absorptions with maximum wavelength and mainly arising from HOMO to LUMO transition) and the ground state for compounds **2-6** by HSEH1PBE. The electron density differences in Figures 4-6 show that the electron density in peripheral anilino groups decreases, while increase of electron density appears in the inner NG cores in the electronic transition. Similarly, the electron density difference in **2e** (Figure 4e) is more obvious than that in **2a** (Figure 4a), attributed to the stronger donating ability of 4-N,N-dimethyl anilino group in **2e**.

Figure 4. The electron density differences between the first excitation state and the ground state of **2** (a, **2a**, b, **2b**, c, **2c**, d, **2d** and e, **2e**). (Blue and red refer to a decrease and an increase in electron density, respectively)

Figure 5. The electron density differences between the first excitation state and the ground state of **4** (a, **4a**, b, **4b** and c, **4c**). (Blue and red refer to a decrease and an increase in electron density, respectively)

Figure 6. The electron density differences between the first excitation state and the ground state of **6** (a, **6a** and b, **6b**). (Blue and red refer to a decrease and an increase in electron density, respectively)

According to the reviewer's comment on "the choice of DFT functional and various functionals may provide different frontier orbitals.", we compared the electronic structure of **2a** calculated by B3LYP, CAM-B3LYP, M062X and HSEH1PBE. Nearly the same frontier molecular orbitals (HOMO-1, HOMO, LUMO and LUMO+1) were revealed by these four hybrid functionals (Figure 7).

Figure 7. Selected key frontier molecular orbitals of **2a** calculated by four different functionals.

The discussion and details about calculations are more detailed in the revised manuscript and Supplementary Information.

Figure 2 – How was the “Absorption” defined? Details about acquiring solid state UV/Vis should be provided at least in ESI.

Response: The “absorption” in the figure 2 was measured in a diffuse-reflectance mode on a UV-Vis-NIR spectrometer (Cary 5000 spectrometer).

The samples for solid state absorption were prepared by mixing and grinding the crystals of NGs and BaSO₄ powder. Using BaSO₄ powder as the blank, the absorption spectra of the sample were obtained automatically by the spectrometer.

The absorption was defined in the caption of Figure 2 in the text and the details on acquiring solid state UV-Vis spectra were provided in Supplementary Information.

Reviewer #2 (Remarks to the Author):

This manuscript has presented synthesis of a series of Donor-Acceptor molecular architectures by amination of perchlorinated nanographenes with anilines, and structures and properties of these molecules, as well as the complexes of these molecules with TTF exhibiting intermolecular charge transfer. These D-A molecular architectures are interesting functional π -molecules with potential applications as organic electronic materials and functional dyes. However, two key findings as claimed in this manuscript have not been supported with solid experimental evidences.

1) The reactions of perchlorinated nanographenes 1, 3 and 5 with anilines by palladium-catalyzed Buchwald–Hartwig are claimed as regioselective, and only the products that have the vertex chlorine atoms substituted by nitrogen were reported. However, the yields of compounds 2a-e are 26-32%, the yields of 4a-c are 13-20%, and the yields of 6a-b are as low as 8-19%. What are the other products (68% to 92%) of these reactions? Did amination always occur on the vertex chlorines?

Response: We appreciate the reviewer for the valuable comments. We carefully investigated the yield, byproducts and selectivity of the C-N coupling reaction between perchlorinated NGs and anilines, taking **2a** as a typical case.

These D-A NGs had a strong absorption on silica, which resulted in the tailing and absorption during the chromatographic separation by silica column. As shown in thin layer chromatography (TLC) analysis, besides the major component (2 spot 2 shown in TLC) which we collected as **2a**, the subsequent eluent (spot 3 shown in TLC) contained **2a** as well (Figure 8). The silica column also absorbed a lot of products after separation

(Figure 8, right). The yield of **2a** (31%) was calculated using the mass of pure component and the less pure components were neglected (Figure 8).

Figure 8. TLC analysis of crude products (spot 1), major component collected as **2a** (spot 2), subsequent eluent (spot 3), isolatable byproduct (spot 4) and the photo of silica column after separation. The byproduct (spot 4) was obtained by additional chromatographic separation.

Therefore, we tried to measure the yield of **2a** by NMR. The detailed procedures were listed below.

100 mg compound **1** reacted with aniline under the reported conditions. After the reaction, the mixture was extracted by dichloromethane (DCM). Then the organic phase was washed with water, dried by MgSO_4 and filtered over silica (100-200 mesh). After evaporation of the solvent, the products were dissolved by 4 ml DCM. Then 20ml methanol was added to precipitate the products. The orange precipitate was collected as the crude products (94 mg) by filtration.

Note that the yield based on the amount of orange crude products was about 70% (the theoretical yield is 130 mg), which meant 30% mass was lost after pretreatments. Because palladium can catalyze the C-C homocoupling of aryl halides under similar conditions as well (*Chem. Rev.* **2002**, *102*, 1359-1469; *J. Org. Chem.* **2006**, *71*, 1284-1287), we speculated that C-C homocoupling between **1** acted as a competitive reaction for the C-N coupling between **1** and aniline. The C-C homocoupling between **1** would lead to the oligomerization and produce less soluble or insoluble oligomers, which were

easy to be removed by pretreatments, such as extraction and filtration. This speculation was supported by the decreased yield of **2a** (15% and 18%) as well as the amount of orange crude products after pretreatments (65 and 80 mg) when the amount of aniline (1 and 1.5 fold of aniline according to the molar of chlorines at the vertexes of **1**) was reduced, in comparison with our reported conditions (4 fold of aniline) (See Table S1 in Supplementary Information).

TLC analysis of orange crude products (spot 1 in TLC, Figure 8) showed one dominant spot, indicating **2a** as the major product. We used NMR spectroscopy to determine the content of **2a** in the orange crude products directly, instead of chromatographic separation. The mass content of **2a** in the crude products by NMR is ~70%, which corresponded to an overall yield of ~50%. Then the crude products were subjected to chromatographic separation as reported, yielding 39 mg of pure **2a**, which shows the similar yield (30%). The less pure components were collected and purified by chromatography again, which afforded additional 10 mg **2a** and 15 mg byproduct (spot 4 shown in TLC, Figure 8). Therefore, we can obtain 49 mg **2a** in total and the isolated yield can be improved to 38%. The lower isolatable yield of **2a** than that determined by NMR was attributed to the strong absorption on silica.

However, considering that six C-N bonds are formed in our reactions, the reported yields (32%-8%) required 83% to 65% yields for each C-N bond formation, that is acceptable for the C-N coupling for aryl chloride. If we considered the yield of **2a** measured by NMR (~50%), the yield for each C-N bond formation is about 89%.

According to the comment “Did amination always occur on the vertex chlorines?”, the byproduct was analyzed by mass spectroscopy (Figure 9) and assigned as pentakis-anilino products [$C_{42}Cl_{13}(NHC_6H_5)_5$ and $C_{42}Cl_{12}(NHC_6H_5)_5O$]. Products with more than 6 anilino groups were not observed.

Figure 9. Mass spectrum of byproduct of **2a** (4 shown in TLC analysis) acquired on a Bruker microflex LRF MALDI-TOF mass spectrometer using tetracyanoquinodimethane (TCNQ) as the matrix in cation mode. The experimental and calculated spectra are represented in red, blue and purple, respectively.

Moreover, the reported experimental conditions required a large excess of aniline (4 fold of aniline) to favor the multiple C-N coupling of **1** with aniline and suppress the homocoupling between **1**. We performed control experiments with different amounts of aniline (1, 1.5, 4, 15 and 30 fold). We found the decreased amount of aniline resulted in a lower yield of **2a** (15% for 1 fold and 18% for 1.5 fold), whereas the experiments with 15 and 30 fold excess of aniline did not affect the yield (30%-31%) (Table S1). The extreme excess of aniline did not affect the yield of the reaction, which validated the inactivity of chlorines in the bay.

In short, we can identify ~70% of the products as the remaining absorbed **2a** (~10-20%), byproducts with less than 6 anilino groups (~10-15%), and some oligomers from the C-C homocoupling (~30%). Taking the isolatable byproduct and yields of control experiments together, the regioselective reactivity of chlorines at the vertexes was confirmed.

The related discussions on the yield, byproducts and control experiments were added in the revised Supplementary Information.

2) The structures of compounds 2a, 2c, 2d, 2e, 4a and 4c were unambiguously characterized by X-ray crystallography. It is reasonable to establish the structures of 2b and 4b on the basis of these crystal structures and comparison of ^1H NMR spectra of 2b and 4b with those of 2a, 2c, 2d, 2e, 4a and 4c. However, the claimed structures of 6a and 6b lack solid support from experimental results. On Page 5 of the Supplementary Information, the authors noted the difficulties in conducting elemental analysis and mass spectra. However, without molecular formula as determined by elemental analysis or mass spectra, one cannot even know how many chlorine atoms in compound 5 were substituted with anilines. Without appropriate experimental evidences, the structures of 6a and 6b should not be claimed.

Response: Thanks for the valuable comments. The complementary experiments on the structure of compound **6** were carried out. ^{13}C NMR spectrum of **6a** was obtained on a 600M Bruker NMR spectrometer equipped with a ultra-sensitive cryo-probe. The ^{13}C NMR spectrum of **6a** was acquired and accumulated for 12 hours. As shown in Figure 10, there are 30 and 8 ^{13}C peaks in the arene and alkane region, respectively, which confirmed the expected structure of **6a**. However, the insufficient solubility of **6b** did not allow the ^{13}C NMR characterization.

Figure 10. ^{13}C NMR spectrum of **6a** in $\text{C}_2\text{D}_2\text{Cl}_4$. ^{13}C NMR (151 MHz, $\text{C}_2\text{D}_2\text{Cl}_4$) δ 139.88, 139.87, 136.93, 136.92, 136.63, 135.74, 135.47, 133.04, 132.89, 128.39, 128.38, 128.10, 128.07, 125.16, 124.71, 124.44, 124.32, 123.85, 123.53, 123.37, 123.01, 122.87, 119.58, 119.46, 119.41, 118.87, 118.53, 117.39, 116.53, 116.23, 34.15, 34.08, 32.88, 32.84, 21.71, 21.65, 13.24, 13.20 ppm. The peaks at 28.87 ppm was assigned to the signal of hexane in the solvent, marked by asterisk.

We did great effort on the mass spectroscopic characterizations of these D-A NGs and found that their mass spectra highly depended on the experimental conditions of MALDI-TOF mass spectrometers, even using the same matrix.

The high resolution mass spectra of **2**, **4** and **6** acquired by AB SCIEX 5800 MALDI-TOF mass spectrometer (AB Sciex Pte. Ltd. USA) normally showed a lot of unassignable fragments, although the molecular ion peak can be found (Figures 11-13). However, **2e**, **4b**, **4c** and **6b** could show clearer high resolution mass spectra (Figures 11-13).

Figure 11. High resolution mass spectra of **2** acquired on a AB SCIEX 5800 MALDI-TOF mass spectrometer using TCNQ as the matrix in cation mode. The experimental and calculated spectra were represented in red and blue, respectively.

Figure 12. High resolution mass spectra of **4** acquired on a AB SCIEX 5800 MALDI-TOF mass spectrometer using TCNQ as the matrix in cation mode. The experimental and calculated spectra were represented in red and blue, respectively.

Figure 13. High resolution mass spectra of **6** acquired on a AB SCIEX 5800 MALDI-TOF mass spectrometer using TCNQ as the matrix in cation mode. The experimental and calculated spectra were represented in red and blue, respectively.

This time, we tried a Bruker microflex LRF MALDI-TOF mass spectrometer (Bruker Corporation), which had lower resolution, to measure the mass spectra of these compounds. We can obtain clear mass spectra for these compounds (Figures 14-16). The byproduct of **2a** was also analyzed by a Bruker microflex LRF MALDI-TOF mass spectrometer (Figure 9). Though the resolution of the spectra acquired by Bruker microflex LRF MALDI-TOF is lower, the structural formulae of **2**, **4** and **6** can be assigned.

Figure 14. Mass spectra of **2** acquired on a Bruker microflex LRF MALDI-TOF mass spectrometer using TCNQ as the matrix in cation mode. The experimental and calculated spectra were represented in red and blue, respectively.

Figure 15. Mass spectra of **4** acquired on a Bruker microflex LRF MALDI-TOF mass spectrometer using TCNQ as matrix in cation mode. The experimental and calculated spectra were represented in red and blue, respectively.

Figure 16. Mass spectra of **6** acquired on a Bruker microflex LRF MALDI-TOF mass spectrometer using TCNQ as matrix in cation mode. The experimental and calculated spectra were represented in red and blue, respectively.

As a supplement, we also tried other softer ionization sources for mass spectroscopy to avoid fragmentation and found that electrospray ionization (ESI) worked for compounds **2a-2d** and **4a-4c** (Figures 17 and 18) as carried out on a Bruker Esquire HCT mass spectrometer (Bruker Corporation). Unfortunately, compounds **2e**, **6a** and **6b** cannot be measured by ESI under the same conditions.

Figure 17. Mass spectra of **2a-2d** acquired on a Bruker Esquire HCT mass spectrometer using ESI in anion mode. The experimental and calculated spectra were represented in red and blue, respectively.

Figure 18. Mass spectra of **4a-4c** acquired on a Bruker Esquire HCT mass spectrometer using ESI in anion mode. The experimental and calculated spectra were represented in red and blue, respectively.

In the revised manuscript, we added the mass spectra of **2a-2d** and **4a-4c** measured by ESI and the mass spectra of **2**, **4** and **6** measured by the Bruker microflex LRF MALDI-TOF mass spectrometer in the Supplementary Information.

Since the mass spectra of **6a** and **6b**, ^1H NMR spectra of **6a** and **6b** and ^{13}C NMR spectrum of **6a** matched the expected structures, we can validate the structure of **6** as was shown in the manuscript.

In order to improve our manuscript, all the ^{13}C NMR spectra of D-A NGs (except **6b**) were provided in the Supplementary Information.

Moreover, some typos and grammar mistakes are found in the manuscript, and the writing should be improved. For example, in Line 34 (Page 2), "a few of" should be "a few", and in Line 78 (Page 5), "substituted" should be "substituting". And the exact meaning of the sentence "The unobstructed chlorines in the bays are staggered" in Line 79 (Page 5) is not clear.

Response: According to the reviewer's suggestion, we have corrected the wording. For clarity, the sentence "The unobstructed chlorines in the bays are staggered" in Line 79

(Page 5) is corrected into “The remaining chlorines in the bays adopt an up and down conformation”.

Therefore, this referee does not recommend this manuscript in the current form to be published in Nature Communications. But a revised manuscript with the above issues addressed appropriately can be reconsidered.

Reviewer #3 (Remarks to the Author):

The authors had previously developed a method to perchlorinate HBCs. In this work, they showed regioselective amination at the vertexes of the perchlorinated HBCs. Taking advantage of the reactivity difference of Cl at the vertex and bay sites, they were able to replace all vertex Cl in reasonable yield. They have also shown that the perchlorinated HBCs have an electron-deficient concave core, which could form charge-transfer supramolecular structures with electron-rich donor like TTF. This review will primarily focus on the chemistry part of the work. The manuscript can be accepted for publication, provided that the evaluation on the charge transfer part of the work is deemed significant.

The regioselectivity demonstrated in this work is quite interesting. It takes advantage of the difference in the steric environment to modulate the reactivity of Cl at the vertex and bay sites in the C-N cross-coupling reaction. The identified reagents and reaction conditions allowed them to completely replace 6 vertex chlorine with aniline. This work will be of interest to synthetic organic chemists and materials scientists working on carbon materials.

How general is this method? The 3 examples have the same number of “layers”, which gave 6 vertex sites for substitution. What are the challenges when the structures are larger?

Response: Many thanks for the valuable suggestions. The concept on regioselectivity of chlorines at the vertexes can be extended to other chemical modifications. Recently, we succeeded in the regioselective thiolation of **1** with 2-thiophenethiol by nucleophilic substitution, producing the hexakis-thiophenethiolated chlorinated hexa-peri-hexabenzocoronene (Figure 1). The systematic and detailed studies on the selective C-S coupling at the vertexes are under way and we would like to report this in due course.

Figure 1. Regioselective thiolation at the vertices (a) and crystal structure of hexakis-thiophenethiolated chlorinated hexa-peri-hexabenzocoronene (b). DMI = 1,3-dimethyl-2-imidazolidinone.

We tried the selective C-N coupling with aniline even for the larger NGs, such as $C_{96}Cl_{27}H_3$, which contained 9 chlorine atoms at the vertices. We found that it could react with aniline but did not afford a well-defined product at present. The largest challenge when the structures become larger is that both high reactivity and selectivity are required by the multiple functionalization whereby the larger NGs are normally less reactive and less soluble. Taking our reactions as an example, considering that six C-N bonds are formed in our reactions, the reported overall yields here (32%-8%) required 83% to 65% yields for each C-N bond formation. Another technical challenge for the larger NGs is the difficulty of purification, originating from their decreased solubility and increased irreversible absorption.

In the rest of the ~70% product, what is the distribution of the various substituted products?

Response: We appreciate the reviewer for the valuable comments. We carefully investigated the yield, byproducts and selectivity of the C-N coupling reaction between perchlorinated NGs and anilines, taking **2a** as a typical case.

These D-A NGs had a strong absorption on silica, which resulted in the tailing and absorption during the chromatographic separation by silica column. As shown in thin layer chromatography (TLC) analysis, besides the major component (2 spot 2 shown in TLC) which we collected as **2a**, the subsequent eluent (spot 3 shown in TLC) contained **2a** as well (Figure 8). The silica column also absorbed a lot of products after separation (Figure 8, right). The yield of **2a** (31%) was calculated using the mass of pure component and the less pure components were neglected (Figure 8).

Figure 8. TLC analysis of crude products (spot 1), major component collected as **2a** (spot 2), subsequent eluent (spot 3), isolatable byproduct (spot 4) and the photo of silica column after separation. The byproduct (spot 4) was obtained by additional chromatographic separation.

Therefore, we tried to measure the yield of **2a** by NMR. The detailed procedures were listed below.

100 mg compound **1** reacted with aniline under the reported conditions. After the reaction, the mixture was extracted by dichloromethane (DCM). Then the organic phase was washed with water, dried by MgSO_4 and filtered over silica (100-200 mesh). After evaporation of the solvent, the products were dissolved by 4 ml DCM. Then 20ml methanol was added to precipitate the products. The orange precipitate was collected as the crude products (94 mg) by filtration.

Note that the yield based on the amount of orange crude products was about 70% (the theoretical yield is 130 mg), which meant 30% mass was lost after pretreatments.

Because palladium can catalyze the C-C homocoupling of aryl halides under similar conditions as well (*Chem. Rev.* **2002**, *102*, 1359-1469; *J. Org. Chem.* **2006**, *71*, 1284-1287), we speculated that C-C homocoupling between **1** acted as a competitive reaction for the C-N coupling between **1** and aniline. The C-C homocoupling between **1** would lead to the oligomerization and produce less soluble or insoluble oligomers, which were easy to be removed by pretreatments, such as extraction and filtration. This speculation was supported by the decreased yield of **2a** (15% and 18%) as well as the amount of orange crude products after pretreatments (65 and 80 mg) when the amount of aniline (1 and 1.5 fold of aniline according to the molar of chlorines at the vertexes of **1**) was reduced, in comparison with our reported conditions (4 fold of aniline) (See Table S1 in Supplementary Information).

TLC analysis of orange crude products (spot 1 in TLC, Figure 8) showed one dominant spot, indicating **2a** as the major product. We used NMR spectroscopy to determine the content of **2a** in the orange crude products directly, instead of chromatographic separation. The mass content of **2a** in the crude products by NMR is ~70%, which corresponded to an overall yield of ~50%. Then the crude products were subjected to chromatographic separation as reported, yielding 39 mg of pure **2a**, which shows the similar yield (30%). The less pure components were collected and purified by chromatography again, which afforded additional 10 mg **2a** and 15 mg byproduct (spot 4 shown in TLC, Figure 8). Therefore, we can obtain 49 mg **2a** in total and the isolated yield can be improved to 38%. The lower isolatable yield of **2a** than that determined by NMR was attributed to the strong absorption on silica.

According to the comment “what is the distribution of the various substituted products?”, the byproduct was analyzed by mass spectroscopy (Figure 9) and assigned as pentakis-anilino products [$C_{42}Cl_{13}(NHC_6H_5)_5$ and $C_{42}Cl_{12}(NHC_6H_5)_5O$]. Products with more than 6 anilino groups were not observed.

Figure 9. Mass spectrum of byproduct of **2a** (4 shown in TLC analysis) acquired on a Bruker microflex LRF MALDI-TOF mass spectrometer using tetracyanoquinodimethane (TCNQ) as the matrix in cation mode. The experimental and calculated spectra are represented in red, blue and purple, respectively.

Moreover, the reported experimental conditions required a large excess of aniline (4 fold of aniline) to favor the multiple C-N coupling of **1** with aniline and suppress the homocoupling between **1**. We performed control experiments with different amounts of aniline (1, 1.5, 4, 15 and 30 fold). We found the decreased amount of aniline resulted in a lower yield of **2a** (15% for 1 fold and 18% for 1.5 fold), whereas the experiments with 15 and 30 fold excess of aniline did not affect the yield (30%-31%) (Table S1). The extreme excess of aniline did not affect the yield of the reaction, which validated the inactivity of chlorines in the bay.

In short, we can identify ~70% of the products as the remaining absorbed **2a** (~10-20%), byproducts with less than 6 anilino groups (~10-15%), and some oligomers from the C-C homocoupling (~30%). Taking the isolatable byproduct and yields of control experiments together, the regioselective reactivity of chlorines at the vertexes was confirmed.

The related discussions on the yield, byproducts and control experiments were added in the revised Supplementary Information.

The charge transfer complex was done on the H or alkyl substituted aniline. How would other substituents on aniline (eg. OCH₃) impact the complex formation and their properties?

Response: The charge transfer complex of hexakis-4-methoxy-anilino chlorinated C₆₀ (**4c**) and tetrathiafulvalene (TTF) was obtained and characterized by X-ray diffraction (Figure 19). The supramolecular complex exhibited a charge transfer peak at 770 nm (Figure 19), which is similar to that of supramolecular complex between **4a** and TTF.

Figure 19. Structure of 2·**4c**⊃TTF and UV-Vis-NIR spectra of **4c** and 2·**4c**⊃TTF in the solid state.

The hexakis-4-methoxy-anilino chlorinated hexa-peri-hexabenzocoronene (**2d**) cannot form the supramolecular complex with TTF by slow solvent evaporation. The synthesis for the supramolecular complex of **2d** and TTF remains to be explored. The hexakis-4-methoxy-anilino chlorinated C₇₈ (**6b**) can form the complex with TTF, but the structure must be confirmed due to the small size of the crystals (~10 μm) of the complex.

The solid state UV-Vis-NIR spectrum of the complex 2·**4c**⊃TTF was added in the Supplementary Information.

Several different terminologies were used to describe their materials, for example, HBC, nanographene, larger graphene, graphene nanoribbons. One should be very careful choosing the correct description of these materials, as the term “graphene” refers to a very special type of material having unique physical properties.

Response: We agree with the comments of reviewer. These terminologies were carefully used in the revision, especially for the term “graphene”.

The reaction conditions in Figures 1 and 3 are missing the critical aniline compound.

Response: Thanks very much for the comments. The reaction conditions were added in Figures 1 and 3.

The quality of the spectra could be improved. Several spectra have solvent peaks, silicone grease in addition to water peaks.

Response: Thank you very much for the comments. The spectra were improved and the ^{13}C NMR spectra of these compounds were also provided in Supplementary Information.

Compound labels are missing in several figures. For example, compound 1 in Figure 1A, compound 3 in Figure 3A, compound 5 in Figure 3B.

Response: Thanks for the comments. The compound labels were added in Figures 1 and 3.

Abbreviations should be defined for the benefit of general audience, for example, $\text{Pd}_2(\text{dba})_3$, rac-BINAP, TTF.

Response: Thanks for the comments. The abbreviations for $\text{Pd}_2(\text{dba})_3$, rac-BINAP, TTF were defined in the revised manuscript.

Reviewers' comments:

Reviewer #1 (Remarks to the Author):

The authors very carefully addressed and answered all my comments. The manuscript reports on important and innovative research, which deserves publication in Nat. Commun.

Reviewer #2 (Remarks to the Author):

Clearly, the authors tried to address the questions raised by this reviewer. Unfortunately, they have not appropriately addressed all the questions.

1. The authors did try to isolate and analyze the by-products. But they failed to demonstrate that the byproducts with less than 6 anilino groups have only chlorine atoms at the vertex positions substituted by nitrogen atoms. Without evidences to support substitution of vertex chlorine atoms, the palladium-catalyzed Buchwald–Hartwig reaction cannot be claimed as regioselective.

2. The extra experiments did not provide sufficient evidences to support the structures of structures of 6a and 6b. They authors may report the palladium-catalyzed reaction of 5, but shall not claim the structures of 6a and.

Therefore, this referee does not recommend this manuscript in the current form to be published in Nature Communications. But a revised manuscript with the above issues addressed appropriately can be reconsidered.

Reviewer #3 (Remarks to the Author):

The authors have taken the time and efforts to address the questions raised by reviewers. The responses are satisfactory.

Reviewers' comments:

Reviewer #1 (Remarks to the Author):

The authors very carefully addressed and answered all my comments. The manuscript reports on important and innovative research, which deserves publication in Nat. Commun.

Response: We appreciate the kind consideration of our manuscript for publication in Nature Communications.

Reviewer #2 (Remarks to the Author):

Clearly, the authors tried to address the questions raised by this reviewer. Unfortunately, they have not appropriately addressed all the questions.

1. The authors did try to isolate and analyze the by-products. But they failed to demonstrate that the byproducts with less than 6 anilino groups have only chlorine atoms at the vertex positions substituted by nitrogen atoms. Without evidences to support substitution of vertex chlorine atoms, the palladium-catalyzed Buchwald–Hartwig reaction cannot be claimed as regioselective.

Response: We appreciate the reviewer for the valuable comments. We tried to get unambiguous structure of byproducts by single-crystal X-ray diffraction, however, a series of trials to grow the single crystals of byproducts, such as vapor evaporation, vapor diffusion and solvent diffusion, were failed. The difficulty of crystal growth for the byproduct we got was ascribed to its low purity. As shown in its mass spectrum, there are two different molecules found.

Figure 1. Mass spectrum of byproduct of **2a** acquired on a Bruker microflex LRF MALDI-TOF mass spectrometer using tetracyanoquinodimethane as the matrix in cation

mode. The experimental and calculated spectra are represented in red, blue and purple, respectively. Obviously, two molecules can be assigned (M_1 and M_2) in the spectrum.

Therefore, we can not unambiguously demonstrate the substitution pattern of the byproducts and withdraw the claim on the regioselectivity of palladium-catalyzed Buchwald–Hartwig reaction as the reviewer suggested. In the revised manuscript, we deleted all the claims on regioselectivity of the reaction and only described the isolated and characterized compounds **2** and **4** showing a selective amination at the vertexes.

2. The extra experiments did not provide sufficient evidences to support the structures of structures of **6a** and **6b**. They authors may report the palladium-catalyzed reaction of **5**, but shall not claim the structures of **6a** and.

Response: Thanks for the reviewer's the valuable comments. The single crystals of **6b** can be obtained but it preferred to form fiber-shaped crystals with a diameter around 10 μm , which is too small for the X-ray diffraction even using the synchrotron X-ray beam. The present experimental evidences of **6a** and **6b** based on mass spectra and NMR spectra can conclude the chemical formula as $\text{C}_{78}\text{Cl}_{20}(\text{NHC}_6\text{H}_4\text{R})_6$ ($\text{R} = \text{C}_4\text{H}_9$ for **6a** and OCH_3 for **6b**). Therefore, we follow the suggestion of reviewer to remove detailed substitution pattern of **6** and report that the palladium-catalyzed reaction of **5** afforded the hexakis-aminated chlorinated products **6a-6b** $\text{C}_{78}\text{Cl}_{20}(\text{NHC}_6\text{H}_4\text{R})_6$ ($\text{R} = \text{C}_4\text{H}_9$ for **6a** and OCH_3 for **6b**) in the revision.

Therefore, this referee does not recommend this manuscript in the current form to be published in Nature Communications. But a revised manuscript with the above issues addressed appropriately can be reconsidered.

Reviewer #3 (Remarks to the Author):

The authors have taken the time and efforts to address the questions raised by reviewers. The responses are satisfactory.

Response: Thank you very much for the kind consideration of our manuscript for publication in Nature Communications.

REVIEWERS' COMMENTS:

Reviewer #2 (Remarks to the Author):

In the revised manuscript, the authors have addressed all the questions raised by this reviewer. This manuscript in its current form is recommended for publication in Nature Communications.

REVIEWERS' COMMENTS:

Reviewer #2 (Remarks to the Author):

In the revised manuscript, the authors have addressed all the questions raised by this reviewer. This manuscript in its current form is recommended for publication in Nature Communications.

Response: Thank you very much for your recommendation for publication in Nature Communications.